# NK Cell Subset Redistribution and Antibody Dependent Activation after Ebola Vaccination in Africans

**DOI:** 10.3390/vaccines10060884

**Published:** 2022-05-31

**Authors:** Helen R. Wagstaffe, Omu Anzala, Hannah Kibuuka, Zacchaeus Anywaine, Sodiomon B. Sirima, Rodolphe Thiébaut, Laura Richert, Yves Levy, Christine Lacabaratz, Viki Bockstal, Kerstin Luhn, Macaya Douoguih, Martin R. Goodier

**Affiliations:** 1Department of Infection Biology, London School of Hygiene and Tropical Medicine, London WC1E 7HT, UK; h.wagstaffe@imperial.ac.uk; 2Department of Infectious Disease, Imperial College London, London W2 1PG, UK; 3KAVI—Institute of Clinical Research University of Nairobi, Nairobi 19676, Kenya; oanzala@uonbi.ac.ke; 4Makerere University—Walter Reed Project, Kampala 16524, Uganda; hkibuuka@muwrp.org; 5Medical Research Council/Uganda Virus Research Institute and London School of Hygiene and Tropical Medicine Uganda Research Unit, Entebbe P.O. Box 49, Uganda; zacchaeus.anywaine@lshtm.ac.uk; 6Centre National de Recherche et de Formation sur le Paludisme (CNRFP), Unité de Recherche Clinique de Banfora, 1487 Avenue Kumda Yonré, Ouagadougou 01 BP 2208, Burkina Faso; s.sirima.cnlp@fasonet.bf; 7Bordeaux Population Health Research Center, University Bordeaux, Inserm, UMR 1219, 33000 Bordeaux, France; rodolphe.thiebaut@u-bordeaux.fr (R.T.); laura.richert@u-bordeaux.fr (L.R.); 8CIC 1401, EUCLID/F-CRIN Clinical Trials Platform, F-33000 Bordeaux, France; 9Inria SISTM Team, F-33405 Talence, France; 10Inserm U955, Vaccine Research Institute, Université Paris-Est Créteil, Hôpital Henri Mondor, 94010 Creteil, France; yves.levy@inserm.fr (Y.L.); christine.lacabaratz@inserm.fr (C.L.); 11Janssen Vaccines and Prevention, 2333 CP Leiden, The Netherlands; vbockstal@gmail.com (V.B.); kluhn@its.jnj.com (K.L.); mdouogui@its.jnj.com (M.D.); 12Flow Cytometry and Immunology Platform, MRC Unit the Gambia at London School of Hygiene and Tropical Medicine, Banjul P.O. Box 273, The Gambia

**Keywords:** Ebola, vaccine, natural killer cell, antibody, Africa

## Abstract

Natural killer cells play an important role in the control of viral infections both by regulating acquired immune responses and as potent innate or antibody-mediated cytotoxic effector cells. NK cells have been implicated in control of Ebola virus infections and our previous studies in European trial participants have demonstrated durable activation, proliferation and antibody-dependent NK cell activation after heterologous two-dose Ebola vaccination with adenovirus type 26.ZEBOV followed by modified vaccinia Ankara-BN-Filo. Regional variation in immunity and environmental exposure to pathogens, in particular human cytomegalovirus, have profound impacts on NK cell functional capacity. We therefore assessed the NK cell phenotype and function in African trial participants with universal exposure to HCMV. We demonstrate a significant redistribution of NK cell subsets after vaccine dose two, involving the enrichment of less differentiated CD56^dim^CD57^−^ and CD56^dim^FcεR1γ^+^ (canonical) cells and the increased proliferation of these subsets. Sera taken after vaccine dose two support robust antibody-dependent NK cell activation in a standard NK cell readout; these responses correlate strongly with the concentration of anti-Ebola glycoprotein specific antibodies. These sera also promote comparable IFN-γ production in autologous NK cells taken at baseline and post-vaccine dose two. However, degranulation responses of post-vaccination NK cells were reduced compared to baseline NK cells and these effects could not be directly attributed to alterations in NK cell phenotype after vaccination. These studies demonstrate consistent changes in NK cell phenotypic composition and robust antibody-dependent NK cell function and reveal novel characteristics of these responses after heterologous two dose Ebola vaccination in African individuals.

## 1. Introduction

The control of Ebola virus outbreaks continues to present a significant ongoing challenge for the African continent, with cases being confirmed in the Democratic Republic of the Congo as recently as April 2022 [1]. Adenovirus type 26 (Ad26).ZEBOV and modified vaccinia Ankara (MVA)-BN-Filo is a safe and immunogenic, two dose anti-Ebola vaccine regimen which has undergone phase 1 and 2 clinical trials in Europe and Africa and marketing authorization approval under exceptional circumstances by the European Medicines Agency (EMA) [2,3,4,5,6,7,8]. Exploratory investigations of the immune responses induced after both dose one (Ad26.ZEBOV) and dose two (MVA-BN-Filo) of this vaccine are of importance for the further understanding of the immune response to vaccination.

Vaccine-induced anti-Ebola antibodies induce NK cell antibody-dependent cellular cytotoxicity (ADCC) in human PBMC in vitro and antibody-dependent cellular mechanisms are associated with post-exposure therapeutic protection in animal models [9,10,11,12].

Our previous studies show that serum collected post-dose one and post-dose two from Ad26.ZEBOV, MVA-BN-Filo vaccinated individuals induced significant NK cell degranulation (surface expression of CD107a), higher IFN-γ secretion and CD16 (FcγRIII) downregulation compared with baseline (pre-vaccination) serum [13]. These studies in European trials also demonstrate that this vaccine induces a redistribution of NK cell subsets towards less differentiated CD56^bright^ and CD57^−^ NK cells accompanied by increased frequency of CD25^+^ NK cells, suggesting potential for increased responsiveness to CD4^+^ T cell derived IL-2 (CD25 expression) and proliferation 21 days after the second vaccine dose [14]. Further, we demonstrated persistent redistribution of NK cell subsets up to 180 days post-dose 2, also involving an enrichment of CD56^bright^ NK cells and enhanced CD25 expression [15].

There is substantial evidence that the functional differentiation of innate and acquired immune cells differs considerably between geographically distinct populations, [16] and that advanced differentiation of both NK cells and CD8^+^ T cells in African populations is associated with a high prevalence of human cytomegalovirus (HCMV) infection [17,18]. Regions with high endemicity for HCMV infection coincide with those targeted for prophylactic vaccination against Ebola infection. HCMV infected individuals have dominant expansions of NK cells expressing the c-type lectin-like receptor NKG2C and lacking the adaptor protein FcεR1γ [19,20]. Our recent studies indicate that antibody-dependent NK cell activation after heterologous two dose Ebola vaccination is influenced by these expansions in HCMV seropositive Europeans [15].

In the present study, ex vivo NK cell phenotypic changes and in vitro anti-EBOV glycoprotein (GP) antibody-dependent NK cell responses were evaluated in samples from a phase 2 clinical trial of the Ad26.ZEBOV, MVA-BN-Filo vaccine at sites in Burkina Faso, Kenya and Uganda, where HCMV exposure is ubiquitous [7,21,22]. Two vaccination strategies were assessed with varying time intervals (28 or 56 days) between dose one (Ad26.ZEBOV) and dose two (MVA-BN-Filo).

NK cell responses may contribute to durable immune responses of target populations receiving the Ad26.ZEBOV, MVA-BN-Filo vaccine.

## 2. Materials and Methods

### 2.1. Study Participants and Samples

Eligible, healthy adult volunteers were recruited into a multi-centre, randomised, placebo-controlled, observer blind Ebola vaccine trial, conducted in several African countries; EBL2002 (ClinicalTrials.gov Identifier registered 29 September 2015, NCT02564523 https://clinicaltrials.gov/ct2/show/record/NCT02564523?term=EBL2002&cond=Ebola+Virus+Disease&draw=2&rank=2, accessed on 14 March 2022). Active vaccination comprised monovalent Ad26.ZEBOV encoding the GP of the Ebola Zaire virus (Mayinga variant) followed by multivalent MVA-BN-Filo encoding of the GP of the Sudan and Zaire Ebola viruses and Marburg virus together with Tai Forest virus nucleoprotein (Janssen Vaccines and Prevention B.V., The Netherlands and Bavarian Nordic, Denmark). Groups one and two received Ad26.ZEBOV on day 1 and MVA-BN-Filo, respectively, on day 29 and day 57.

The EBL2002 trial protocol and study documents were approved by the relevant authorities and ethics committees. For the samples used in this study, approvals were obtained under the principal trial from the Joint Kenyatta National Hospital—University of Nairobi Ethics Research Committee (Ref Number KNH/ERC/R/206; The Uganda Virus Research Institute Research Ethics committee (Ref Number GC/127/18/08/157) and The Institutional Ethics Committee, Centre Muraz, Burkina Faso (Ref Number HB/MGK/2019-002/EBOVAC-2). Serum and PBMC samples were collected at baseline (Visit 0), post-dose one (day of dose two administration), days 29, or 57 (Visit one), 21 days post-dose two (Visit two) and 180 days post-dose two (Visit three). PBMC and serum samples from 29 actively vaccinated adult participants were analysed for this study (Table 1). A predominance of male study samples reflected the distribution of participants in the original clinical trial [7]. In some experiments, a total of seven study participants from placebo arms (Group one, *n* = 3; Group two, *n* = 4) were analysed in parallel. Ex vivo phenotyping and ADNKA assays with a standard NK cell readout were performed on samples from all 29 actively vaccinated participants and all seven placebo control individuals. For NK cell activation with autologous sera, a subset of 18 paired PBMC and serum samples were used for each study visit. A blood sample from a standard healthy unvaccinated donor was obtained in the UK for standard NK cell readout assays with approval from the Ethics committee of The London School of Hygiene & Tropical Medicine (LSHTM, reference number 14383). The current study was approved by the ethics committee of London School of Hygiene and Tropical medicine (Ref numbers 14760-2). All participants of the EBL2002 trial provided written informed consent.

PBMC from EBL2002 volunteers from Burkina Faso, Kenya and Uganda were isolated using Leucosep tubes cryopreserved in liquid nitrogen. PBMC separation and cryopreservation at each study site were performed under controlled conditions and samples validated on receipt by Janssen Vaccines for storage. For this study PBMC samples were then shipped to LSHTM in liquid nitrogen vapour phase and sera were shipped on dry ice. EBOV GP-specific IgG concentration was determined by EBOV GP Filovirus Animal Non-Clinical Group (FANG) ELISA [23]. All study volunteers involved in this study were tested seropositive for HCMV as determined by IgG ELISA (Demeditec, Kassel, Germany).

### 2.2. NK Cell Assays

Ex vivo phenotypic analysis or antibody-dependent NK cell activation (ADNKA) assays were performed after recovery of cryopreserved PBMC [15]. Recovered PBMC were stained for ex vivo flow cytometric analysis or reserved for ADNKA assay in RPMI 1640 supplemented as previously described [15]. ADNKA assays were performed in 96 well flat bottom plates pre-coated with recombinant Ebola virus GP (EBOV GP; Zaire strain, prepared in Hek293F cells; (Stratech, Ely, UK) as previously described [15]. PBMC from a single, non-study donor (non-vaccinated; from fresh blood) or from vaccinated EBL2002 trial participants (cryopreserved) were seeded (3 × 10^5^/well) onto the antigen-coated plates together with 5% pre- or post-vaccination sera (autologous in the case of vaccine study PBMC) or with 5% Visit 2 serum pooled from all donors at and incubated for 6 h at 37 °C. Positive control cultures comprised the CD20 expressing human Burkitt’s Lymphoma cell line (RAJI; ECACC, Salisbury, UK) used at a PBMC: target cell ratio of 5:1 with monoclonal Rituximab anti-CD20 at a final concentration of 100 ng/mL (Rituxan^®^, Genentech, San Francisco, CA, USA). GolgiPlug (Brefeldin A; 1/1000 final concentration; BD Biosciences, Wokingham, UK) and GolgiStop (Monensin; 1/1500 concentration; BD Biosciences) were added for the final 3 h of culture. Cells were harvested into round bottom plates for staining by soaking and resuspension in cold PBS (with 0.5% FCS, 0.05% sodium azide and 2 mM EDTA).

### 2.3. Flow Cytometry

Anti-CD107a-FITC (clone H4A3; BD Biosciences) was added to the cultures (2 µL per well) for the entire culture period. Cells were stained for all other surface markers including a viability marker (Fixable Viability Stain 700; BD Biosciences) in FACS buffer (PBS, 0.5% FCS, 0.05% sodium azide and 2 mM EDTA) for 30 min in 96-well round bottom plates after blocking Fc receptors for 5 min with Fc Receptor (FcR) Blocking Reagent (Miltenyi Biotec, Woking, UK). Cells were then washed in wash buffer, fixed and permeabilized using Cytofix/Cytoperm Kit (BD Biosciences) or Foxp3/Transcription Factor Fixation/Permeabilization Kit (ThermoFisher, Altricham, UK) according to the manufacturer’s instructions. Cells were then stained for intracellular markers with FcR blocking for 20 min and washed again. Finally, cells were resuspended in FACS buffer and analysed using a BD LSRII flow cytometer (Becton Dickinson, Oxford, UK). Cells were acquired using FACSDiva software and data were analysed using FlowJo V10 (FlowJo LLC, Beckton Dickinson). FACS gates were set using unstimulated cells or fluorescence minus one (FMO) controls. Samples with less than 100 NK cell events were excluded from the analysis (less than 3% of samples).

Ex vivo phenotypic and in vitro phenotypic analysis of NK cells was performed separately after labelling with the following reagents: anti-CD3-V500 (clone UCHT1) (BD Biosciences), anti-CD56-BV605 (clone HCD56), anti-CD16-APC (clone CB16), anti-CD57-e450 (clone TB01), anti-Ki67-PerCP-eFluor710 (clone 20Raj1), anti-NKG2C-PE (clone 134591) (Biotechne, Abingdon, UK), anti-FcεR1γ^−^FITC (rabbit polyclonal Ig) (Millipore, UK), anti-CD25-BV785 (clone BC96) (Biolegend, London, UK) and anti-IFN-γ-BV785 (clone 4S.B3).

### 2.4. Statistics

A statistical analysis was performed using GraphPad Prism version 7.04 (Dotmatics, Boston, MA, USA). Functional responses were compared using Wilcoxon signed-rank test or one-way ANOVA mixed effects analysis with correction for multiple comparisons (detailed in figure legends). Correlation of two variables was determined using a Pearson’s correlation analysis. Significance levels are assigned as * *p* ≤ 0.05, ** *p* ≤ 0.01, *** *p* ≤ 0.001, and **** *p* ≤ 0.0001 for all tests.

## 3. Results

### 3.1. Longitudinal NK Cell Subset Changes after Heterologous 2 Dose Ebola Vaccination

We performed a phenotypic assessment of NK cells from baseline (Visit zero, Day one), after the first vaccine dose (Visit one, Day 29 or Day 57), 21 days post-dose 2 (Visit two, Day 50 or day 78) and 180 days after dose two (Visit three) (see Appendix A for NK cell and subsets analysis gating strategies). Initially, we pooled data from all actively vaccinated participants from both vaccine strategies for analysis (see Table 1 for participant characteristics).

We observed no significant change in the frequencies of total NK, CD56^bright^ and CD56^dim^ NK cells or in the overall proliferation (Ki67^+^) of these subsets between visits (Figure 1A,B,E,F). However, after sequential gating of CD57 or FcεR1γ defined subsets within CD56^dim^ NK cells, we observed a significant increase at visit two in the proportion of ‘canonical’ FcεR1γ^+^ NK cells with a reciprocal reduction in the representation of ‘adaptive’ FcεR1γ^−^ NK cells and a trend towards an increase in the proportion of less differentiated CD57^−^ cells (Figure 1C,D) post-vaccination. These changes were accompanied by significant increases in the proliferation of both CD56^dim^CD57^−^ and CD56^dim^FcεR1γ^+^ cell subsets (Figure 1G,H). Overall, CD56^dim^CD57^+^ or CD56^dim^FcεR1γ^−^ NK cell subsets exhibited lower frequencies of proliferating cells compared to CD56^dim^CD57^−^ and CD56^dim^FcεR1γ^+^, respectively, and these were unaffected by vaccination (Figure 1G,H), in contrast with our previous studies.

In Europeans [15], no significant changes in NK cell subset distributions were observed 180 days after vaccine dose two (Visit three) as compared to baseline (Visit zero) (Figure 1C,D,G,H). Furthermore, we did not see any alterations in the frequencies of NK cells or subsets expressing CD25 (IL-2Ra) in these African samples over time (data not shown). There was no significant change in the frequencies of highly differentiated (CD57^+^) or adaptive (FcεR1γ^−^) NK cells expressing the HCMV associated c-type lectin-like receptor, NKG2C (Figure 1I,J). No significant change is the distribution of NK cell subsets or in the proliferation of these was observed in study participants from the placebo group (Appendix A).

We then analysed the ex vivo NK cell subset distribution, proliferation and activation responses according to a 28 day (Group one) or 56 day (Group two) interval occurring between vaccine doses (Appendix A). No significant change in the frequencies or proliferation of CD56^dim^, CD56^bright^ or CD57-defined NK cell subsets was seen (Appendix A). However, a shorter dosing interval (Group 1, 28 days) resulted in a significant increase in the frequency of CD56^dim^FcεR1γ^+^ NK cells at visits one and two, whereas no significant effect was observed for a longer period between vaccine doses (Group two, 56 days) (Appendix A). This increase was consistent with increased proliferation within the CD56^dim^FcεR1γ^+^ subset in Group one (Appendix A).

### 3.2. Post-Ad26.ZEBOV, MVA-BN-Filo Vaccination-Induced Antibody-Dependent NK Cell Activation at 21 Days Post-Dose 2

We measured antibody-dependent NK cell activation (ADNKA) in response to baseline (Visit zero), post-dose one (Visit zero), 21 days post-dose two (Visit two) serum plus immobilized EBOV GP using a uniform responder NK cell preparation (PBMC from a single blood donor) (Figure 2). Frequencies of CD107a and IFN-γ expressing NK cells were analysed in addition to the geometric mean fluorescence intensity of CD16, which is down regulated upon antibody-dependent NK cell activation (flow cytometric analysis for a representative study participant is shown in Appendix A).

When data from both vaccination groups were combined for analysis, there was a significant induction of CD107a and IFN-γ expression, and a significant down-regulation of CD16 expression mean fluorescence intensity (MFI) on CD56^dim^ NK cells in response to stimulation with EBOV GP in the presence of 21 days post-dose two (Visit two) serum compared with baseline serum (Figure 2A–C). No significant changes in ADNKA responses were observed using sera from individuals within the placebo arm (Appendix A). Comparing ADNKA responses between vaccine groups, no impact of dosing interval was observed on the frequency of degranulating CD56^dim^ NK cells or on CD16 expression (Figure 2D,E) post-vaccination. Significantly higher frequencies of IFN-γ producing CD56^dim^ NK cells were observed only after a 56-day dosing interval (Group two V3, mean = 2.5, S.E.M = 0.46, *p* = 0.019) compared to baseline values (Group two V1, mean = 0.85, S.E.M = 0.26) and this was significantly higher than that observed after a 28-day dosing interval (Group one V3, mean = 0.92, S.E.M = 0.04). and only Group two (Figure 2F). There was no significant difference in baseline IFN-γ frequencies between groups.

### 3.3. Reduced Ebola Glycoprotein Specific Antibody-Dependent Responses in Post-Vaccination NK Cells

Less highly differentiated NK cells are well known to mount superior responses to inflammatory cytokines, while further differentiation and the emergence of adaptive NK cells is associated with reduced cytokine responsiveness and adaptation towards antibody-dependent activation [17,19,20]. The African individuals studied here were universally HCMV infected and had high frequencies of adaptive NK cells compared to our previous study (CD56^dim^FcεR1γ^−^ = 36.1% IQR% 18.2–58.1 compared to 17.9% IQR 5.3–38.4%). We therefore studied the potential impact of the high frequencies of highly differentiated and adaptive NK cell subsets observed in our African vaccine recipients on Ebola virus GP-specific ADNKA responses. Baseline (Visit 0) or post-dose two (Visit two) NK cells within individual PBMC were cultured in the presence of autologous pre- or post-vaccination (Visit two) sera and the frequencies of degranulating or IFN-γ producing cells and the levels of CD16 on these cells was assessed (Figure 3).

Similar to our observations using a standardised NK cell readout (Figure 2), we observed significant antibody-dependent activation (higher CD107a expression, CD16 downregulation and higher IFN-γ production) in autologous NK cells at baseline (Visit zero cells) in the presence of serum taken 21 days post-dose two (Visit two) compared to baseline serum (Figure 3A–C, left columns). We also observed a similar pattern when cells taken at Visit two were assessed using autologous baseline or visit two sera (Figure 3A–C right columns). However, degranulation (CD107a) responses were significantly lower in cells taken at Visit two compared with baseline cells (Figure 3A, right columns vs. left columns). Similar response patterns of NK cells to autologous Visit two sera from individual study donors were observed using pooled sera from all donors tested (V2P), suggesting that the variation in responses could be attributed to individual level differences in NK cell functional capacity (Figure 3A–C).

NK cells from baseline (V0) or post vaccine dose two mounted significant responses to positive control stimulus Rituximab (anti-CD20) crosslinked onto the Raji B tumour cell line (R + R) compared to the negative isotype control (R + I). Notably, no differences were observed between the responses of NK cells taken at Visit zero or Visit two when stimulated with the positive control (R + R, Figure 3D–F). These data suggest that whilst individual level variation in ADNKA may be attributable to intrinsic differences in NK cell functional capacity both before and after vaccination, degranulation responses of post-vaccination NK cells to autologous or pooled Ebola GP specific post-vaccination serum could be influenced by vaccine-induced changes in the NK cell compartment.

A comparison of responses between different vaccination groups suggests that the interval between vaccine doses had no significant impact on ADNKA of baseline or post-vaccination autologous NK cells cultured either with EBOV GP and post-vaccination serum (Appendix A) or with control stimuli (Appendix A). However, consistent with observations across all study participants tested (Figure 3), the frequency of CD107a^+^ cells was significantly lower in group two (*p* = 0.027) amongst CD56^dim^ NK cells taken after vaccination (Visit two) compared with baseline (Visit zero) and tended towards a reduction in group one (*p* = 0.078) (Appendix A). These data again suggest that the dosing interval did not significantly affect ADNKA responses.

### 3.4. HCMV Associated NK Cell Expansions do Not Influence Antibody-Dependent Responses to Ebolavirus Glycoprotein

Adaptive expansions of NK cells are associated with HCMV infection and these cells are reported to mount superior antibody-dependent degranulation responses in a number of experimental systems [19,20]. To further explore whether high frequencies of these HCMV associated adaptive NK cell expansions observed before and after vaccination in this African study population influenced ADNKA, we compared responses in groups of individuals with higher or lower frequencies of adaptive expansions of CD56^dim^FcεR1γ^−^, CD56^dim^CD57^+^NKG2C^+^ NK cells or CD56^dim^FcεR1γ^−^NKG2C^+^ NK cells (Figure 4). Comparing ADNKA responses, we observed no significant difference in the frequencies of CD107a^+^ and IFN-γ^+^ NK cells or in the MFI for CD16 in individuals with low frequencies (lower 50th centile) of adaptive NK cell populations compared to those with larger (upper 50th centile) expansions (Figure 4).

Similar outcomes were observed after stratification for CD56^dim^FcεR1γ^−^ (Figure 4A–C), CD56^dim^CD57^+^NKG2C^+^ (Figure 4D–F) or CD56^dim^FcεR1γ^−^NKG2C^+^ NK cell frequencies (Figure 4G–I). Degranulation responses were significantly lower when autologous cells taken post-dose two (Visit 2) were stimulated in the presence of matched serum compared to cells taken at baseline (Visit zero), irrespective of the frequencies of adaptive NK cell subsets (Figure 4A,D,G). Overall, these data suggest that the adaptive expansions of NK cells do not significantly influence the magnitude of ADNKA responses and that mechanisms independent of NK cell functional differentiation could account for reduced degranulation responses in post-vaccination NK cells.

### 3.5. Anti-Ebolavirus GP Antibody Concentration does not Correlate with the ADNKA Response of Autologous NK Cells

ADNKA responses of a standardised readout in response to EBOV GP and multiple serum samples from Visit two correlated significantly with the anti-EBOV GP antibody concentration determined for each sample used (Figure 5, upper panels). CD16 MFI was significantly negatively correlated with antibody concentration (r = −0.612, *p* = 0.0004. Figure 5B) and CD107a and IFN-γ showed significant positive correlations (r = 0.495, *p* = 0.006 and r = 0.658, *p* = 0.0001, respectively. Figure 5A,C). Ebola GP-specific ADNKA responses of NK cells collected from individual participants taken at baseline (Figure 5D–F) or post-dose two (Figure 5G–I) did not correlate with the antibody concentration determined for each serum sample. These data suggest that intrinsic features of host NK cells combine with antibody associated factors, including concentration, to determine the individual ADNKA response. We demonstrate distinct changes in NK cell phenotype and functional characteristics in this study cohort after heterologous two dose Ebola vaccination with Ad26.ZEBOV administration followed by MVA-BN-Filo which differ from those previously observed in European studies.

## 4. Discussion

Compared with previous studies in Europeans of this and other Ebola vaccines [15,24], we saw no change in the least differentiated CD56^bright^ NK cell subset, whereas higher frequencies and increased proliferation of cells with canonical, intermediate differentiation phenotype (CD56^dim^CD57^−^ or CD56^dim^FcεR1γ^+^) were observed 21 days after vaccine dose two. CD56^bright^ NK cells are reduced in number and frequency within the first few years of life in Africans, whereas these are maintained at higher frequencies throughout life in European individuals [17]. These characteristics and a preponderance of highly differentiated and adaptive NK cells in Africans could therefore account for differences in vaccine-induced effects. The effects on CD56^bright^ NK cells in European cohorts are observed in response to a number of different vaccines through heightened responsiveness to cytokines IL-12, IL-18 and IL-2, [25,26,27,28] although these responses were not evident in African studies using equivalent vaccines [29,30].

Vaccine induced proliferation of less differentiated cells observed ex vivo may reflect responsiveness to cytokines induced either by the vaccine or by the vector. Interestingly, an interval of 28 days between vaccine dose one and vaccine dose two resulted in more significant effects on the frequencies of CD56^dim^FcεR1γ^+^ NK cells than the longer interval of 56 days, possibly reflecting a reduction in responsiveness to the second dose. Previous studies after a single dose influenza vaccination demonstrated heightened responsiveness of NK cells to restimulation in vitro was lost within 90 days after vaccination, potentially reflecting transient epigenetic effects on less differentiated NK cells which affect cytokine responsiveness.

In this study we also demonstrate robust ADNKA responses after Ad26.ZEBOV, MVA-BN-Filo vaccination both in a standardised NK cell assay and in autologous baseline and post-vaccination NK cells up to 21 days after vaccine dose two. We observed no significant impact of dosing interval on the antibody-dependent NK cell degranulation (CD107a) response. However, a longer dosing interval of 56 days induced a higher frequency of IFN-γ producing cells in the ADNKA assay, potentially reflecting stronger boosting of antibody levels in some individuals in this group and consistent with increased immunogenicity after a delayed second dose. This is in line with the benefits of delaying the second dose on immunogenicity in other trials of this two dose Ad26.ZEBOV, MVA-BN-Filo vaccine regimen [2,3,6,8].

Expansions of adaptive NK cells (defined by the expression of NKG2C, activation of killer cell immunoglobulin-like receptors and by a loss of FcεR1γ) are observed during acute HCMV infection and persist lifelong during periods of viral latency or reactivation [17,19,20,25,31]. These HCMV associated adaptive NK cell subsets express higher levels of CD16, the low affinity Ig Fc receptor and stable epigenetic changes promote responsiveness to antibody-dependent signals. In contrast with our previous study where 59% of individuals were HCMV infected with low HCMV overall IgG titres (median 70.8, IQR 35.1–12.3) [15], the African individuals studied here were universally HCMV infected with relatively high antibody titres (Table 1). Correspondingly, African study participants had high frequencies of adaptive NK cells compared to our previous European study [15]. However, while our previous observations were that NKG2C^+^FcεR1γ^−^cell frequencies influenced the overall IFN-g response in a European cohort, we observed no difference in the frequency of cells mounting CD107a or IFN-γ ADNKA responses when comparing African individuals with lower or higher frequencies of HCMV associated adaptive expansions.

HCMV could potentially affect ADNKA responses through effects on vaccine induced antibodies. However, in contrast with other studies on influenza vaccination in adults or tetanus, specific antibody responses after diphtheria-tetanus-whole cell pertussis vaccination in infants [32,33,34], we observed no association between anti-HCMV antibody titres and vaccine induced anti-EBOV GP antibody concentration in this study (data not shown). A recent study found no significant impact of HCMV associated T cell or NK cell differentiation phenotype on the immunogenicity of chimpanzee adenovirus vectored coronavirus vaccine [35]. It is likely that a complex interplay influences vaccine induced antibody responses, involving factors such as age and the relative timing of HCMV infection and Ebola vaccination. Additionally, we cannot rule out the possibility of interactions with other chronic infections, including malaria and helminth parasites which are the focus of further studies.

Effective ANDKA responses in both baseline and post-vaccination NK cells indicate that a balanced response is achieved involving both canonical and adaptive NK cells. Cytokines induced by Ebola glycoprotein could conceivably co-stimulate the responses of less differentiated canonical NK cells in our assay system and could also contribute to occasional in vitro IFN-γ responses observed in placebo controls. Ebola virus glycoprotein induces a number of NK cell activating pro-inflammatory cytokines, including IL-18, which is known to contribute to innate NK cell activation and to co-stimulate ADNKA responses to pathogens and vaccine antigens [14,36,37].

We observed significantly lower degranulation responses in NK cells post-dose two compared to baseline NK cells when combined with sera from matched participants. Paradoxically, no significant difference in CD16 downregulation or IFN-γ production was observed in baseline and post-vaccination NK cells and there was no difference in NK cell function in response to control stimuli, indicating these NK cells had similar intrinsic function, irrespective of vaccination time point. In addition to NK cell activating cytokines, Ebola virus glycoprotein induces a number of anti-inflammatory cytokines, including IL-10, which are also elevated during infection [14,38] and which could potentially suppress the degranulation response of post-vaccination NK cells within our PBMC cultures. Antibody Fc mediated effects have been associated with vaccine induced protection against Ebola virus in non-human primates, although not directly involving NK cells [39]. Maturation of the antibody response, including the emergence of isotypes which bind to distinct Fc receptors for antibodies on NK cells could also potentially contribute to the modulation of the ADNKA degranulation response after vaccine dose two. The significance of reduced antibody-dependent NK cell degranulation responses while maintaining NK cell IFN-γ after vaccination on vaccine-induced immunity remains to be elucidated.

In summary, this study demonstrates transient alteration to NK cell differentiation and proliferation and robust antibody-dependent NK cell responses to Ad26.ZEBOV, MVA-BN-Filo Ebola virus vaccination in Africa cohorts. This study also highlights features of the NK cell phenotype and effector function which contributes to our understanding of vaccine-induced immune response in populations with differing levels of exposure to infectious agents.

## Figures and Tables

**Figure 1 vaccines-10-00884-f001:**
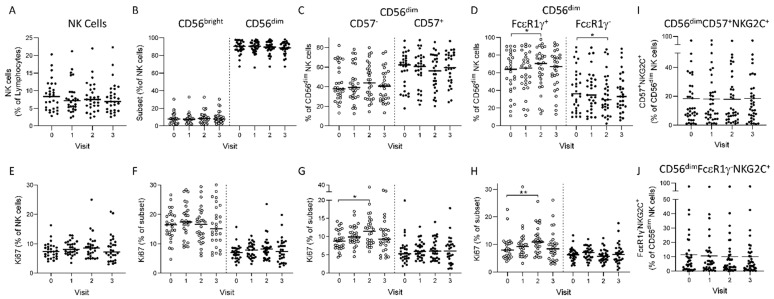
Ex vivo analysis of NK cell subsets and activation markers. Frequencies of NK cells and subsets at Visit zero (baseline), Visit one (post-vaccine dose one), Visit two (21 days post vaccine dose two) and Visit three (180 days post-vaccine dose two) (**A**) NK cells (**B**) CD56^bright^ and CD56^dim^ (**C**) CD56^dim^CD57^−^ and CD57^+^ and (**D**) CD56^dim^FcεR1γ^+^ and FcεR1γ^−^
**(E)** Ki67 expression within NK cells and within (**F**) CD56^bright^ and CD56^dim^ (**G**) CD56^dim^CD57^−^ and CD57^+^ and (**H**) CD56^dim^FcεR1γ^+^ and FcεR1γ^−^ NK cell subsets (**I**) Frequencies of CD57^+^NKG2C^+^ and (**J**) FcεR1γ^−^NKG2C^+^ subsets within CD56^dim^ NK cells across the vaccination time course. Vaccination groups were pooled for analysis and all actively vaccinated individuals are represented (*n* = 29). Paired comparisons between time points were made using Wilcoxon signed-rank test. * *p* < 0.05, ** *p* < 0.01.

**Figure 2 vaccines-10-00884-f002:**
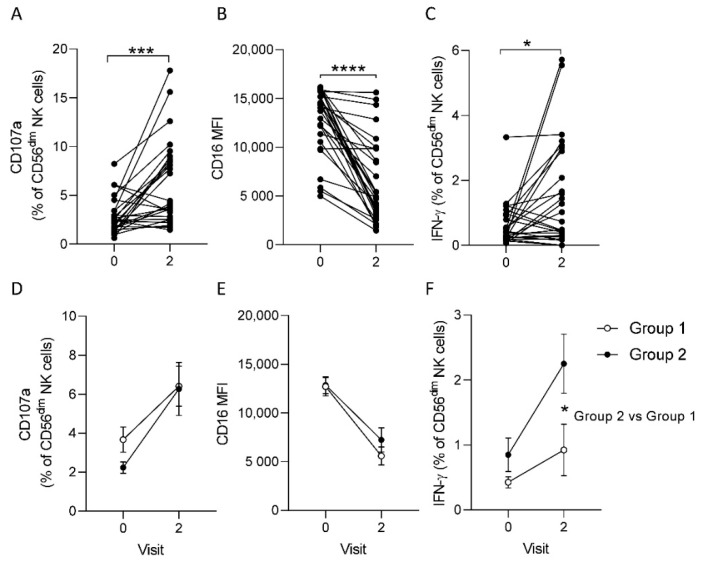
Antibody-dependent NK cell responses to immobilized EBOV GP. CD107a (**A**,**D**), CD16 MFI (**B**,**E**) and IFN-γ (**C**,**F**) expression within gated CD56^dim^ NK cells from a standard PBMC readout (from a single unvaccinated donor) cultured with serum samples from individual trial donors collected pre-vaccination (Visit zero) and 21 days post-dose two (Visit 2). Data are presented for combined vaccination groups (**A**–**C**) and for individual vaccination group (**D**–**F**) (see Table 1 for participant numbers in each group). Plots show individual data points before and after vaccination (**A**–**C**) and mean values with standard error of the mean (s.e.m.) for vaccination group one (open symbols) and group two (filled symbols) (**D**–**F**). Across visit (**A**–**C**) and intergroup (**D**–**F**) comparisons were performed using one-way ANOVA mixed effects analysis with Geisser-Greenhouse correction. * *p* < 0.05. *** *p* < 0.001. **** *p* < 0.0001.

**Figure 3 vaccines-10-00884-f003:**
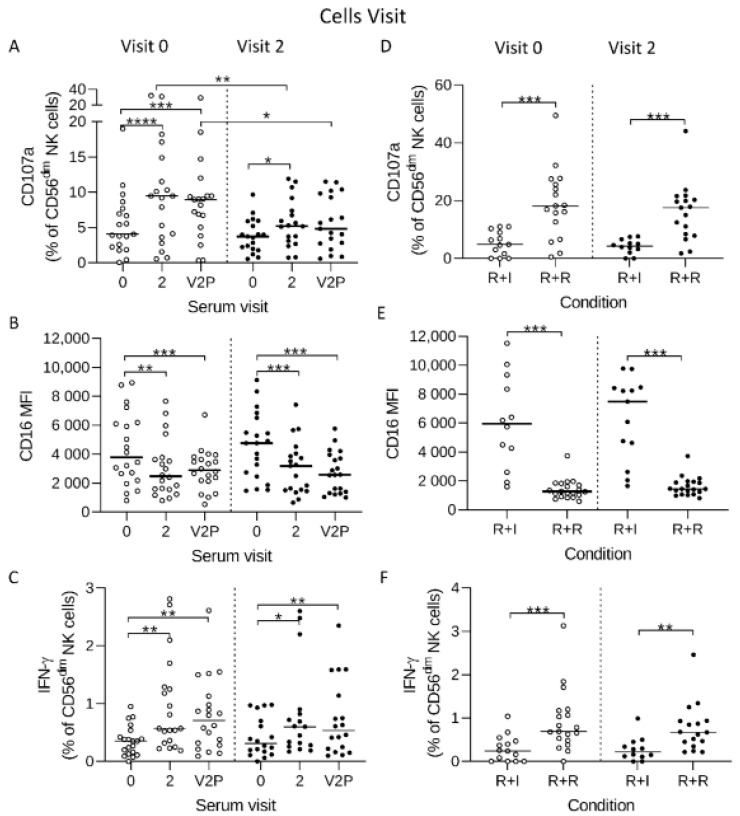
ADNKA responses of pre- and post-vaccination NK cells from individual trial participants using their autologous pre- and post-vaccination sera. PBMC collected pre-vaccination (Visit 0) and 21 days post-dose two (Visit two) were cultured with EBOV GP in the presence of autologous serum from the same visit or with pooled serum from all participants at Visit two (V2P) (**A**–**C**). Positive and negative control stimuli were Raji B cells + Rituximab™ (R + R condition) or isotype control antibody (R + I condition), respectively (**D**–**F**). Responses were analysed by flow cytometry for (**A**,**D**) CD107a, (**B**,**E**), CD16 mean fluorescence intensity (MFI) and (**C**,**F**) IFN-γ expression. Individual data points are shown for 18 participants tested in this assay with a line representing median values for all tested donors. Comparisons between visits were performed using one-way ANOVA mixed effects analysis with Geisser-Greenhouse correction. * *p* < 0.05, ** *p* < 0.01, *** *p* < 0.001. **** *p* < 0.0001.

**Figure 4 vaccines-10-00884-f004:**
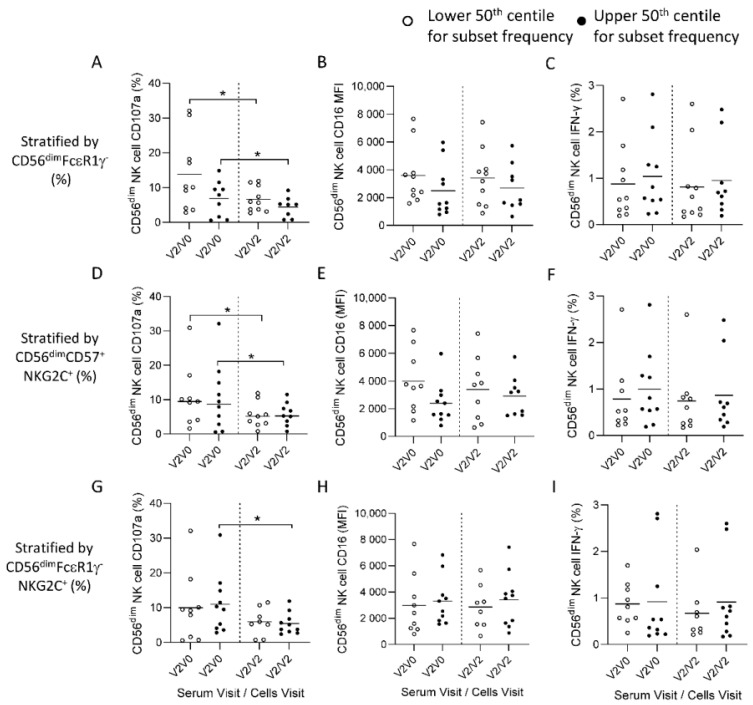
HCMV-associated adaptive expansions and ADNKA responses. Comparison of ADNKA cell CD107a (**A**,**D**,**G**), CD16 mean fluorescence intensity (MFI) (**B**,**E**,**H**) and IFN-γ (**C**,**F**,**I**) expression in groups defined according to HCMV associated NK cell expansions. Responses to autologous post vaccination serum (21 days post-dose two; Visit two) combined with either baseline (V0) or post-dose two (V2) cells were assessed (**A**–**C**). Responses were compared between lower (*n* = 9, open circles) and upper (*n* = 9, filled circles) 50th centiles for frequencies of CD56^diim^FcεR1γ^−^ (**A**–**C**); CD56^dim^CD57^+^NKG2C^+^; (**D**–**F**) or CD56^dim^FcεR1γ^−^NKG2C^+^ subsets; **G**–**I**) as determined in Figure 1. Data points are shown with a line representing median values. Comparisons across visits were performed using a one-way ANOVA mixed effects analysis with a Geisser-Greenhouse correction. * *p* < 0.05.

**Figure 5 vaccines-10-00884-f005:**
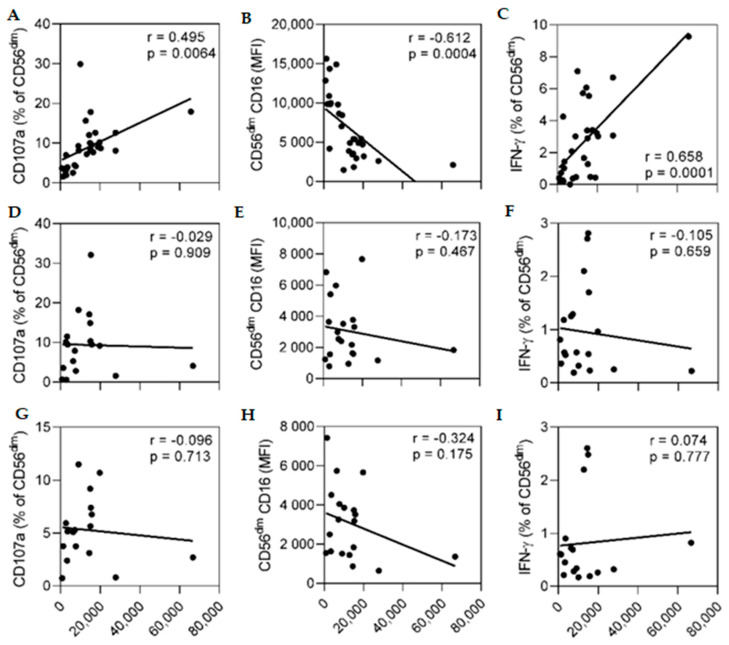
Correlation between vaccination-induced anti-GP antibody concentration and NK cell responses. ADNKA responses in a single standardized donor readout (**A**–**C**) or from 19 individual study participants at Visit zero (**D**–**F**) or Visit two (**G**–**I**) were cultured with EBOV GP in the presence of serum collected from the trial individuals 21 days post-dose two (Visit two). For the standard readout, 27 sera were tested for each study time point. For the autologous readouts, 18 sera were tested for each study time point. NK cell CD107a (**A**,**D**,**G**), CD16 MFI (**B**,**E**,**H**) and IFN-γ (**C**,**F**,**I**) expression was compared against anti-GP antibody concentration (measured at Visit 2) (ELISA units/mL, Table 1). Goodness-of-fit line was determined by linear regression and r- and *p*-value analysis was performed by Pearson’s correlation. Significance was defined as *p* < 0.05.

**Table 1 vaccines-10-00884-t001:** Randomised vaccination regimen and corresponding numbers of serum and PBMC samples used in this study.

	Group 1	Group 2
**Vaccine Schedule**	Ad26 ^1^, MVA ^2^ 28 Day Interval	Ad26, MVA 56 Day Interval
Age years: median (IQR)	32, (24–46)	25, (22–44)
Sex: *n* (m/f)	12/2	10/5
Baseline samples (Visit 0) (PBMC/Serum)	Day 1 (14/14)	Day 1 (15/15)
Post-dose 1 samples (Visit 1) ^3^ (PBMC/Serum)	Day 29 (14/0)	Day 57 (15/0)
21 days post-dose 2 samples (Visit 2) (PBMC/Serum)	Day 50 (14/14)	Day 78 (15/15)
180 days post-dose 2 samples (Visit 3) (PBMC/Serum)	Day 209 (14/0)	Day 237 (15/0)
Anti-Ebola GP IgG (Visit 2) concentration (EU/mL) (median, IQR)	11,513 (3409–16,021)	13,379 (3367–19,788)
Anti-HCMV IgG (Visit 0) concentration (IU/mL) (median, IQR)	2879 (2492–3918)	2677 (1258–4299)

Study participants received monovalent Ad26.ZEBOV encoding the GP of the Ebola Zaire virus (Mayinga variant) followed by multivalent MVA-BN-Filo encoding the GP of the Sudan and Zaire Ebola viruses and Marburg virus together with Tai Forest virus nucleoprotein. Groups one and two received Ad26.ZEBOV on day one (Visit 0) and MVA-BN-Filo on day 29 or 57 respectively (Visit 1). ^1^ Adenovirus type 26.ZEBOV; ^2^ Modified vaccinia Ankara-BN-Filo; ^3^ This time point corresponds to the day of administration of vaccine dose two. PBMC = peripheral blood mononuclear cells; GP = glycoprotein; IgG = Immunoglobulin G; EU/mL = ELISA units per millilitre; HCMV = human cytomegalovirus; IQR = interquartile range.

## Data Availability

The data that support the findings of this study are available from the corresponding author upon reasonable request.

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
