# Peer review of "NK Cell Subset Redistribution and Antibody Dependent Activation after Ebola Vaccination in Africans"

_vaccines, 2022, doi:10.3390/vaccines10060884_

Round 1

Reviewer 1 Report

General comments:
This is a well-written and intriguing paper on the roles of NK cells and ADCC in the potential protection
against Ebola virus disease. The ability to use clinical data, in the context of the population in which
Ebola vaccines are needed most, provides insight into contributions to differences in immunogenicity
profiles across geographic regions. Two overarching comments would help strengthen the manuscript:
1. They role that HCMV plays in differential immune responses. The authors note the ubiquity of
HCMV exposure in African populations. It may be beneficial to include similar data from subjects
fromoneofthetrialsconductedinEurope(ifsuchdataisavailableandhasn’tbeenpublished
elsewhere), focused on HCMV-negative individuals. Even if included in supplemental data to
show the differences, it may be of value.
2. Is there a reason placebo-controlled subjects were not included in the data sets? If I understood
correctly, all comparisons are either from baseline to post-vaccination, or between the day 1/29
and 1/57 regimens. Placebo control data would be a valuable addition.
Specific comments:
• Line51:Seemstobefromthejournal’stemplateinstructions.
• Line 61-74: This is a style comment only, but it may help to flow if that paragraph is separated
into two to first detail what is known about NK cells, ADCC, etc. in terms of protecting against
Ebola. And then speak to what has been observed in the Ad26/MVA clinical trials.
• Table 1: can the authors speak to the reason the sex balance is skewed towards males?
• Figure 1 appears to be missing from the draft.
• Figure 2: The open and closed symbols are confusing as laid out. In A-C it appears to reflect visit
0 (open) and visit 2 (closed). But in D-F, and the in-figure legend applies only to D-F, open and
closed symbols reflect groups 1 and 2. Would suggest just making all symbols closed in A-C. Or
noting individual subjects as open or closed, consistent with D-F.

Author Response

Many thanks your helpful comments. We have addressed these below after each of your and taken them into consideration when revising our manuscript.

This is a well-written and intriguing paper on the roles of NK cells and ADCC in the potential protection against Ebola virus disease. The ability to use clinical data, in the context of the population in which Ebola vaccines are needed most, provides insight into contributions to differences in immunogenicity profiles across geographic regions. Two overarching comments would help strengthen the manuscript:
1. The role that HCMV plays in differential immune responses. The authors note the ubiquity of HCMV exposure in African populations. It may be beneficial to include similar data from subjects from one of the trials conducted in Europe (if such data is available and hasn’t been published elsewhere), focused on HCMV-negative individuals. Even if included in supplemental data to how the differences, it may be of value.

The data from the European trial (with equivalent vaccine regimen and dosing interval) were published in January 2021 (NpJ Vaccines, doi: 10.1038/s41541-021-00280-0.). We have highlighted comparisons of the frequencies of HCMV+ individuals and of HCMV associated NK cell subsets and differences seroprevalence in the text of the results and discussion sections, respectively (lines 293-295 and 457-460 of the track changes version of the manuscript).

  1. Is there a reason placebo-controlled subjects were not included in the data sets? If I understood correctly, all comparisons are either from baseline to post-vaccination, or between the day 1/29and 1/57 regimen.

We initially excluded the placebo control data as we recognised that n=7 provided us with limited power to perform reliable analysis on this group. However, we have now included placebo control data for the full range of phenotypic data. Detail of the placebo control group is given in the methods (lines 130-131 ofthe track hanges version of the manuscript) There were no significant change in phenotype or proliferative capacity for placebo control group over the time course and mentioned this in the text (Figure S2, lines 233-235 of the track changes version of the manuscript). We also include a supplementary figure for antibody dependent NK cell activation using sera from placebo control individuals and a standard NK cell readout (Figure S5, lines 266-267 of the track changes version of the manuscript). No overall significant change in NK cell ADNKA function was observed in the placebo control group, although increased in IFN-γ expression was seen in two individuals which may be attributed to innate effects of ebola glycoprotein (discussed in lines 486-487 of the track changes version of the manuscript).

Specific comments:
• Line 51:Seems to be from the journal’s template instructions.

Many thanks for drawing this to our attention - This line has now been removed from the populated template.

  • Line 61-74: This is a style comment only, but it may help to flow if that paragraph is separated into two to first detail what is known about NK cells, ADCC, etc. in terms of protecting against Ebola. And then speak to what has been observed in the Ad26/MVA clinical trials.

Thank you for this suggestion. We have rearranged this paragraph accordingly (lines 60-78 of the track changes version of the manuscript).

  • Table 1: can the authors speak to the reason the sex balance is skewed towards males?

Our experiments were based on a subset of samples selected randomly from the main EBL2002 clinical trial in African study sites. The predominance male subjects in our sample set reflects to an extent the frequency in the originating trial with 67.6% of male participants in group 1 and 68.3% in group 2. We have noted this in the text of the materials and methods section (page, line 128-130 of the track changes version of the manuscript).

  • Figure 1 appears to be missing from the draft.

This figure is indeed present on page 6 of the journal formatted template manuscript, before line 239 of the track changes version and is included in the uploaded TIFF versions of the figures.

  • Figure 2: The open and closed symbols are confusing as laid out. In A-C it appears to reflect visit 0 (open) and visit 2 (closed). But in D-F, and the in-figure legend applies only to D-F, open and closed symbols reflect groups 1 and 2. Would suggest just making all symbols closed in A-C. Or
    noting individual subjects as open or closed, consistent with D-F.

Thank you for this useful suggestion. We have now reformatted figure 2A-C as closed symbols only and amended the figure legend accordingly (Lines 283-284 of the track changes version of the manuscript).

Reviewer 2 Report

The authors of manuscript entitle "NK cell subset redistribution and antibody dependent activation after Ebola Vaccination in Africans" have done an extensive study on individuals who received Ebola vaccination. The authors designed the study in two different groups where one group received 2 doses of vaccine (Ad26, MVA) 28 days interval and the other group received 2 doses of vaccine (Ad26, MVA) 56 days interval. The authors collected the participants PBMC at different points to check the activation of different NK cells and interferon expression.

 The authors performed the study in Africa where all the participant chosen for the study had universal exposure to HCVM. The results show after 2nd dose of vaccination there were a significant redistribution of NK cells and involving enrichment of less differentiated CD56dimCD57- and CD56dim. 

The authors designed the study nicely with proper control and used a valid and accurate methods to evaluate all the samples. The study very rich in results and the results part explain clearly. The authors discussed all the results clearly with all the supporting evidence. 

I have few comment an suggestion on their study.

  1. The full stop at line 54 (Adenovirus type 26 (Ad26).) it cause misunderstanding of the sentence.
  2. It would be interesting if the authors perform neutralization on the Ebola virus using the participant serum in future study.  

Author Response

Many thanks for your interesting commens which we have addressed below in blue text:

I have few comment and suggestion on their study.

  1. The full stop at line 54 (Adenovirus type 26 (Ad26).) it cause misunderstanding of the sentence.

Thank you for this suggestion and we do understand the point being made. We are, however, unable to modify this formatting as it is the established technical name for the vaccine.

2. It would be interesting if the authors perform neutralization on the Ebola virus using the participant serum in future study.  

Virus neuralisation titres were established in the original manuscript with superior neutralisation observed after a longer dosing interval (Barry et al 2021). It would indeed be interesting to establish whether NK cells participated in the killing of pseudovirus infected target cells.

Kind Regards

Martin R Goodier

Round 2

Reviewer 1 Report

Thank you for addressing the comments, this is an excellent addition to the literature!